# Application of Electrochemical Aptasensors toward Clinical Diagnostics, Food, and Environmental Monitoring: Review

**DOI:** 10.3390/s19245435

**Published:** 2019-12-10

**Authors:** Zhanhong Li, Mona A. Mohamed, A. M. Vinu Mohan, Zhigang Zhu, Vinay Sharma, Geetesh K. Mishra, Rupesh K. Mishra

**Affiliations:** 1School of Medical Instrument and Food Engineering, University of Shanghai for Science and Technology, Shanghai 200093, China; zhli@sspu.edu.cn (Z.L.); zgzhu@sspu.edu.cn (Z.Z.); 2School of Environmental and Materials Engineering, College of Engineering, Shanghai Polytechnic University, Shanghai 201209, China; 3Research Center of Resource Recycling Science and Engineering, Shanghai Polytechnic University, Shanghai 201209, China; 4Pharmaceutical Chemistry Dept., National Organization for Drug Control and Research [NODCAR], 6 Abu Hazem Street, Pyramids Ave, 29, Giza 99999, Egypt; mona7722@aucegypt.edu; 5Electrodics & Electrocatalysis Division, CSIR-Central Electrochemical Research Institute (CECRI), Karaikudi 630003, Tamil Nadu, India; vinumohan756@gmail.com; 6Amity Institute of Biotechnology, Amity University Rajasthan, Jaipur 303002, India; vsharma4@jpr.amity.edu; 7Multiscale Fluid Mechanics Lab, School of Mechanical Engineering, Sungkyunkwan University, Suwon 44-746, Korea

**Keywords:** electrochemical biosensors, aptamers, clinical diagnostic, food analysis, environmental analysis

## Abstract

Aptamers are synthetic bio-receptors of deoxyribonucleic acid (DNA) or ribonucleic acid (RNA) origin selected by the systematic evolution of ligands (SELEX) process that bind a broad range of target analytes with high affinity and specificity. So far, electrochemical biosensors have come up as a simple and sensitive method to utilize aptamers as a bio-recognition element. Numerous aptamer based sensors have been developed for clinical diagnostics, food, and environmental monitoring and several other applications are under development. Aptasensors are capable of extending the limits of current analytical techniques in clinical diagnostics, food, and environmental sample analysis. However, the potential applications of aptamer based electrochemical biosensors are unlimited; current applications are observed in the areas of food toxins, clinical biomarkers, and pesticide detection. This review attempts to enumerate the most representative examples of research progress in aptamer based electrochemical biosensing principles that have been developed in recent years. Additionally, this account will discuss various current developments on aptamer-based sensors toward heavy metal detection, for various cardiac biomarkers, antibiotics detection, and also on how the aptamers can be deployed to couple with antibody-based assays as a hybrid sensing platform. Aptamers can be used in various applications, however, this account will focus on the recent advancements made toward food, environmental, and clinical diagnostic application. This review paper compares various electrochemical aptamer based sensor detection strategies that have been applied so far and used as a state of the art. As illustrated in the literature, aptamers have been utilized extensively for environmental, cancer biomarker, biomedical application, and antibiotic detection and thus have been extensively discussed in this article.

## 1. Introduction

Aptamers can be classified as small nucleic acid ligands of single-strand deoxyribonucleic acid (ssDNA), ribonucleic acid (RNA), and peptide molecules [1]. They can bind to their targets’ molecules including viruses, cells, proteins, peptides, and some small organics [2]. Aptamers are flexible enough to bend themselves into well-defined secondary structures to bind to their targets with noble specificity and affinity. Their binding affinities are highly target-dependent (ranges from pico-molar to nano-molar scales) for diverse protein targets [3]. Aside from such characteristics, they have also demonstrated great promise in sensing applications since aptamers can easily be created by chemical synthesis and can also be readily modified with some functional groups and linkers [4,5]. Furthermore, due to their versatility, aptamers are great samples of functional biological molecules that are chosen in vitro [6]. As a result, technologies associated with aptamers have engrossed immense attention in diverse research communities [7]. Other than the significant sensitivity and specificity that aptamers offer, they also provide a broad range of benefits over other existing molecules in terms of flexibility, cost, and stability.

Such characteristics have significantly favored their application as ultra-selective bio-recognition elements for diverse biotechnology related applications. For instance, one class of aptamers that are coated with magnetic beads are used to purify and concentrate analytes from body-fluids [8]. Additionally, integration of aptamers with liquid-chromatography (LC)-mass spectral analyses of biofluids can make enormous implementations in diverse sections of the diagnostics-related industry [9]. All of these characteristics, especially chemical stability, have reversible thermal-denaturation and are resistant to severe pre-conditioning without losing the bioactivity, making aptamers a major challenge for antibodies [1]. Aptamers recommend numerous recompenses when compared to antibodies; these are purely formed antigen-specific proteins through a biological process. The generation of the aptamer does not involve an immune retort in host animals to acquire them, since they are synthesized chemically through nucleic acid selection. Therefore, sensors that employ aptamers as their bio-recognition components (aptasensors) have found applications in many fields of research [10].

The first biosensing applications of aptamers were introduced in 1996 where optical biosensors based on fluorescently labeled aptamers were developed [11]. However, the integration of aptamer characteristics with those of electrochemical systems [12] that include enhanced sensitivity [13] and selectivity [14,15,16], reconcilable with unique microfabrication techniques [17,18] and capable of integrating nanomaterials [19,20,21,22,23,24,25], inherent miniaturization [26], low cost, disposability [27], low power requirements [28,29], independence of sample turbidity, and point-of-care applications [30,31,32] have made electrochemical aptasensors [33,34,35,36] a terrific candidate for many sensing applications. For the first time in 2004 [37], an aptasensor was introduced. To build the sensor, glucose dehydrogenase-labeled aptamers were used to develop a sandwich-type and amperometric aptasensor. Since then, electrochemical aptasensors have been used widely for health monitoring, food safety, and environmental pollution control [38].

Recently, there is a growing requirement to scrutinize ecological contaminations. The water, air, and food items are the major fatalities of the contaminants that have an impact on individuals as well as animal life. Such contamination may lead to very threatening effects and eventually lead to widespread destruction. The environmental contaminants can be divided into organic and inorganic analytes, maybe a toxin from bacteria or fungi, pharmaceutical products or analytes, a drug, phenolic compound, heavy metals, or pesticides of different categories. Even though many new techniques have been developed and devoted to the detection of clinical analytes, food toxins, allergens, and environmental contaminants, there is still a great interest in the development of small and portable sensing devices or colorimetric assays to analyze these small and large molecules responsible for the contamination. Moreover, highly competent bioassay and decentralized techniques have also been developed. Such methods are traditionally explored assays for quantity and quality tests of important analytes and are sensitive; however, they need expensive instruments and technical people to do the analysis and are additionally unsuitable for decentralized analysis. 

As a fascinating biosensing field and due to progressive advancements, this review paper only examines the recent advanced applications of electrochemical aptasensors in food, environmental, and clinical diagnostics. This review also compares different electrochemical aptasensors detection strategies that have so far been applied.

### 1.1. Aptasensors Electrochemical Detection Strategies

#### 1.1.1. Sandwich Sensors Combining Aptamer and Antibody

This is very emerging way to combine an aptamer with an antibody on the sensor or transducer surface when the secondary aptamer is not available. In such situations, the captured aptamers (may also be an antibody, but most often not) are integrated on the transducer or sensing surface, and secondary antibodies/aptamers are used for signal recording or enlargement. There are some reports on combining aptamers and antibodies in various platforms such as optical sensors, electrochemical, and localized surface plasmon resonance (LSPR)based sensors for the detection of a diverse range of analytes (i.e., proteins or viruses). Yan Kang et al. developed electrochemical sensors based on an antibody and aptamer sandwich for thrombin detection [39]. The aptamer was used as a detection probe and methylene blue was used as an electrochemical redox probe intercalating the probing aptamer without previous labeling. A special electrode interface comprising of nano-gold chitosan was used with far better conductivity and analytical performance. The sensor linearity was in the range of 1–60 nM with a detection limit of 0.5 nM. In another hybrid approach of antibody and aptamer, Guo and Kim presented a sensitive method of protein detection that relies on the peak shift phenomenon using localized surface plasmon resonance (LSPR) generated by the aptamer and antigen-antibody sandwich assay, and thrombin was detected on Au nanorods [40].

A new method for the determination of platelet-derived growth factor-BB (PDGF-BB) was developed using an electrochemical immunosensor with an aptamer-primed, long-strand circular detection probe. A rabbit anti-human PDGF-BB polyclonal antibody was immobilized on the electrode to serve as the capture antibody. The detection probe was synthesized via polymerase extension along with a single-stranded circular plasmid DNA template with a primer headed by the anti-PDGF-B aptamer. In the presence of the analyte, the aptamer-primed circular probe was captured on the electrode via the formation of an antibody/PDGF-BB/aptamer sandwiched complex. The electro-activity indicator methylene blue was adsorbed on the electrode surface via the analyte-sandwiched complex with long-strand circular DNA, thus yielding a strong electrochemical signal for the quantification of PDGF-BB. This strategy allowed for electrochemical detection with enormous signal amplification arising from the long-strand localized circular probe. The oxidation peak current of methylene blue in square wave voltammetry measurements showed a linear dependence on the concentration of PDGF-BB in the range from 50 to 500 ng mL^−1^, with a detection limit as low as 18 pg mL^−1^ [41]. In another work, an aptamer–antibody based sandwich biosensor was developed for lysozyme detection in alcoholic beverages. An aptamer was coupled on a carbon based transducer by covalent binding/click chemistry. A biotin labeled antibody was used for the sandwich assay. By adopting this assay, lysozyme was detected in the range from 5 fM to 5 nM with a limit of detection (LOD) of 4.3 fM. This implies that aptamer-antibody based sandwich assays are promising analytical tools for future exploration [42]. Figure 1 illustrates a schematic of an aptamer–antibody based combined approach for analyte detection.

#### 1.1.2. Electrochemical Impedance Spectroscopy Aptasensors

Electrochemical impedance spectroscopy (EIS) is a very sensitive technique to analyze the sophisticated electrical resistance of an electrochemical interface [43]. EIS is particularly well-suited for the detection of binding events on the electrode interface. After the development of the recognition complex between bio-molecules present at the sensing surface and analyte in the solution, the electrical properties of the recognition interface would be directly or indirectly changed [44]. To build a biosensor, the recognition element is immobilized on a sensing surface. Then, to evaluate the binding process at the electrode surface, different strategies of capacitance or resistance can be applied.

Double-layer capacitance (C_dl_) develops across the electrode and electrolyte interface, and it is where electrochemical charge transfer reactions happen. Consequently, all the components on the interface that include immobilized molecules and solvent molecules have a direct effect on the double-layer capacitance. As a result, this technique measures the analyte where no Faradaic current exists. Another approach is the measurement of charge transfer resistance (R_ct_). Since the charge transfer process is involved, therefore, redox moieties exist in solution to develop such Faradaic impedance.

The application of the EIS detection method avoids the necessity to use labeling electroactive groups. In addition, having most of the proteins non-electroactive, the EIS technique is mainly suitable for large proteins since they bring an obvious change in the interfacial charge transfer resistance (R_ct_) when captured on the electrode surface. Figure 2 represents the labeled and label-free approaches in electrochemical biosensors whereas Figure 3 represents the schematic for the principle of target-promoted changes in charge transfer resistance.

## 2. Electrochemical Aptasensors for Clinical Diagnostic Applications

Since the first discovery of the aptamer sequence, a variety of electrochemical aptasensors have been reported with enormous attention on their possible applications in clinical diagnostics. Moreover, the application of aptamers as bio-receptors has provided remarkable prospects in clinical diagnostic assays. The typical electrochemical techniques being employed for aptasensors in clinical diagnostic applications can be cyclic voltammetry (CV), differential pulse voltammetry (DPV), square wave voltammetry (SWV), EIS, and so on. For CV, DPV, SWV, and chrono-amperometry (CA) techniques, the electrochemical redox probes labeled to aptamers require a conductive medium on an electrode surface to generate the signal response. The first way to achieve this goal is that aptamers should be directly or indirectly connected to redox probes; after the special recognition of the aptamer to the target, the conformational changes occur in the aptamer structure, which will cause the distance to change between the redox probe and the electrode surface, leading to the redox current changing. In another way, the electrochemical redox probe or its precursor is pre-added into the electrolysis solution before testing instead of connecting to the aptamer, then the electrochemical redox current in electrolysis solution will change before or after the aptamer recognizes the detection target specifically, which results from the distant changing of the probe’s electron transfer path caused by the tertiary structure changes of the aptamer, or from the redox probe formation obtained by the precursor reaction. For the EIS technique, the detection principle depends on the change in impedance of the electrode (interfacial charge transfer resistance) or interfacial redox capacitance, which results from the special recognition of aptamers modified on the electrode to target. The following section will give emphasis to the most illustrative examples of recently developed electrochemical aptasensors in clinical biomarker detection.

### 2.1. Application of Aptasensors for Biomarker Detection

The presence of biomarkers in body fluids indicates a biological status or process in various conditions and could provide objective information for clinical diagnosis. Analyses of biomarkers in several samples including blood, saliva, urine, and cerebrospinal fluid (CSF) are mostly utilized in clinical diagnosis for the early-stage screening of several diseased conditions. Electrochemical aptasensors have been successfully utilized for the detection of these biomarkers and several diseased conditions have been successfully detected/screened. Most of the aptamer-based biosensors focused on the detection of biomarkers related to cancer, cardiovascular diseases (CVD), dementia and Alzheimer’s disease (AD). Early detection of CVD and cancer is very important to uplift the survival rate of critically ill patients since they claim the highest death rates worldwide [45,46]. Numerous clinically significant biomarkers are being reported in various body fluids that are the signature markers for cancer and CVD. Moreover, they are directly linked with the multiplicity of many proteins related to cancer and CVD [47]. Similarly, dementia is also a very serious health issue worldwide; it presently affects more than 25 million people and is expected to affect more than 75 million people worldwide within the following 20 years. The greatest reason for dementia in elderly persons is AD in 70% cases, where the outcome is progressive damage of cerebral functions. Clinically, AD is diagnosed by the development of amyloid plaques, neurofibrillary masses in the brain, alongside brain atrophy, inflammation, neuronal, and synaptic loss [48]. The emphasis on blood-related biomarkers for AD has developed significantly within recent years. Recognized biomarkers of AD in CSF like Amyloid β peptide (AβP), Apolipoprotein E (ApoE), and total tau (t-tau) proteins are highly accurate and well-characterized for sensitive and specific detection. However, their sample extraction and clinical applications are challenging [49,50]. Recent years have witnessed various reports on the detection of clinically important biomarkers in serum and other body fluids using electrochemical aptasensors. Most recent and illustrative instances of these reported aptasensors are being conferred here.

#### 2.1.1. Aptasensors for Cancer Biomarkers

Specific probes of biosensors help in identifying probable threat factors by detecting cancer biomarkers in blood serum, plasma, free DNA, or other body fluids. Plenty of current reports have effectively utilized aptasensors for the detection of cancer biomarkers [51]. Prostate-specific antigen (PSA) is an important cancer biomarker for prostate cancer that was typically detected with several antigen-antibody (Ag-Ab) based reactions. Recently, an electrochemical aptasensor was described by Jolly et al. for the detection of PSA. They explored the DNA aptamer sequence as a receptor layer and two diverse immobilization approaches were established. In the first method, immobilization of a thiol based aptamer was done on an Au-electrode by mercaptohexanol, whereas in the second method, an Au-electrode was incubated together with thiol terminated sulfobetaine and mercaptoundecanoic acid. Cross-linking in both methods was carried out utilizing 1-ethyl-3-(-3-dimethylaminopropyl) carbodiimide hydrochloride (EDC)/ N-Hydroxysuccinimide (NHS) chemistry. Reduction in electron transfer resistance for [Fe(CN)_6_]^3-/4-^ signals were recorded in both methods after incubation in PSA solution. In the MUA/sulfobetaine monolayer, PSA was detected at the concentration below 1 ng/mL [52]. In another report, a PSA-specific DNA aptamer was immobilized on the pyrolytic graphite electrode using avidin-biotin surface chemistry. The developed aptasensor exhibited excellent linear response toward PSA within the range from 0.25 to 200 ng/mL with a detection limit of 0.25 ng/mL. The authors have also tested the cross-reactivity of the developed aptasensor toward BSA, hemoglobin, and thrombin. The sensor showed negligible changes in the response toward non-specific biomarkers. The stability of the reported aptasensor was up to 30 days for PSA detection [53]. Avian erythroblastosis oncogene B (HER2) is another important biomarker, mostly as a screen to confirm breast cancer. HER2 is a member of the epidermal growth factor receptor (EGFR or ErbB) family and a transmembrane tyrosine kinase receptor. Elevated levels of HER2 in blood serum indicate the potential risk of breast cancer [54,55]. Chun et al. reported a DNA aptamer-based method for the screening of HER2. They utilized mercaptopropionic acid (MPA) based surface activation to immobilize the amine terminal aptamer sequence on an Au-electrode. Cross-linking was carried out by EDC/NHS chemistry followed by blocking with BSA. Ferri/ferrocyanide was utilized as a redox couple to detect HER2-aptamer complex formation. A linear increment in electron transfer resistance was observed from 10^−5^ to 10^2^ ng/mL with the detection limit of 10^−5^ ng/mL for HER2 detection [56]. Vascular endothelial growth factor (VEGF) is another crucial signaling protein accountable for angiogenesis. VEGF is a well-known biomarker for several ailments involving cancer, retinopathy, and rheumatoid arthritis. Qureshi et al. have reported a method for the development of an aptamer–antibody based capacitive sandwich assay for VEGF-165 in serum samples. This method utilizes the capture of VEGF protein by an anti-VEGF aptamer and antibody. The sensor was functionalized with an anti-VEGF aptamer that first captures the VEGF protein, followed by sandwiching with antibody-conjugated magnetic beads (MB-Abs). Non-Faradaic electrochemical impedance spectroscopy (nFIS) was utilized to measure the change in capacitance at different frequencies (50–300 MHz). The reported method demonstrated improved capacitive signals with increasing concentration of the analyte. Authors have reported a wide detection range (5 pg/mL to 1 ng/mL) for VEGF [57]. Recently, a number of electrochemical aptasensors devoted to numerous cancer biomarkers have come up. Mostly, the choice of functionalization and immobilization of aptamer sequence remains a challenge. Although the developed aptasensors may possibly work as a substituted tool in the early screening of cancer, they are limited by the stability of the aptamer functionalized layers and the probability of surface fouling at the time of clinical sample analysis [58]. Osteopontin (OPN), with a molecular weight between 41–75 kDa, is a protein distributed in all body fluids. Its overexpression is considered as the biomarker of breast cancer. A DNA aptamer for human osteopontin was selected using SELEX. This developed aptamer showed a good response toward OPN with a detection limit of 1.4 nM with good reproducibility and acceptable selectivity [59]. It was the first report on the separation and characterization of a high-affinity DNA aptamer toward human OPN using the SELEX process. The fluorescence assay was utilized to determine the binding affinity of the chosen DNA aptamer. Furthermore, the aptamer was utilized as a biorecognition element toward the progression of a label-free DNA aptasensor for human OPN detection using the electrochemical techniques CV and SWV. The attractive performance of such an aptamer-based sensor was evaluated in vitro using standard solutions made in a buffer solution of pH 7.4 and real human plasma samples. Another work on OPN was developed by the same group. This research effort was made using a simple aptasensor that relies on an electrochemical transducer for the detection of the same molecule; human OPN. The molecule was selected based on a previous study [59]. The RNA aptamer recombinant human OPN (rhOPN) was coupled on a gold electrode fabricated by screen printing and through streptavidin-biotin interaction. To assess the response of analyte-aptamer interaction as well as binding, the CV of an [Fe(CN)_6_]^3–/4–^ redox probe was recorded and the biosensor performance was checked toward human OPN detection. The SWV was also used to compare the sensitivity and pursue the aptamer sensor construction that thereby enabled a comparison between the recorded electrochemical performance after each step. They developed a label-free electrochemical RNA aptasensor to detect human osteopontin with the LOD of 3.7 nM [60]. The major difference between these research works was that the first aptasensor was based on DNA aptamers and the first SELEX process remonstration for OPN, whereas the latter one was an RNA based aptasensor and the redox behavior was monitored through of [Fe(CN)_6_]^3–/4–^ on gold electrodes.

#### 2.1.2. Aptasensors for CVD Biomarkers

CVD is reflected as the most important global health problem, therefore, it was an emergent requirement to develop a sensitive, rapid, and economical sensing device for the screening of CVD biomarkers at an early-stage. Electrochemical aptasensors have played a significant part in early stage screening of CVD to some extent. Over the recent years, various aptasensors have been reported for the rapid and economical detection of multiple cardiac biomarkers. Among the variously reported biomarkers, C-reactive protein (CRP) was the utmost studied biomarker, followed by interleukin-6 (IL-6), interleukin-1 (IL-1), and cardiac troponin I or T (cTnI/T). Several researchers have extensively studied these biomarkers and used them to predict cardiovascular events [46,61].

Recently, a disposable electrochemical aptasensor was developed by Centi et al. for the screening of CRP. The aptasensor was constructed utilizing a sandwich format where an RNA aptamer was coupled to a monoclonal antibody and alkaline phosphatase (AP). After the sandwich assay, the modified magnetic beads were captured by a magnet on the surface of a graphite working electrode. DPV was employed to detect electrochemical signals of this event after the addition of the AP substrate (α-naphthyl-phosphate) and α-naphthol produced during the enzymatic reaction. The LOD and LOQ calculated in CRP free serum were reported to be 0.2 and 6 mg/L, respectively [62]. In another work by Qureshi et al. an aptasensor was reported by exploring nFIS as a detection technique. In this method, Au-interdigitated capacitor electrodes were designed on a SiO_2_ surface for the immobilization of the thiolated aptamer. Upon reaction with a CRP-aptamer complex, signals were generated and resulted in a change in the relative capacitance of gold interdigitated(GID) capacitors with the applied frequency. The reported detection range for CRP was determined within the range of 100–500 pg/mL. It was also reported that the highest affinity of CRP-aptamer binding (Kd) was 1.6 μM/L at a 208 mHz frequency [63]. Most recently, Wang et al. reported an electrochemical aptasensor for the detection of CRP exploring functionalized silica microspheres. RNA aptamers toward a specific recognition of CRP were immobilized on the surface of NP modified Au-electrode via the Au–sulfur affinity. The SWV was employed to record the sensing signals generated upon the interaction between a sandwich structure of aptamer–CRP–immuno probe. The developed aptasensor exhibited a large linear range (0.005 ng/mL to 125 ng/mL) for the detection of CRP with excellent LOD at 0.0017 ng/mL [64]. Furthermore, Tertis et al. have recently described a label-free electrochemical aptasensor for the sensitive recognition of IL-6 in human serum. The aptasensor was based on an SPCE modified with a nanocomposite construction consisting of polypyrrole and AuNPs. IL-6 specific aptamer was immobilized using sulfur–gold bonding and monitored using CV and EIS based techniques. Under optimized conditions, the developed aptasensor showed a good impedimetric response toward the target analyte. The results presented that IL-6 can be detected in an extensive linear range from 1 pg/mL to 15 μg/mL with a LOD of 0.33 pg/mL [65].

#### 2.1.3. Aptasensors for Neurotransmitters and Alzheimer’s Biomarker

Dopamine (DA) is a very important biomarker and crucial neurotransmitter molecule with a diverse function in the central nervous system. It is reported that DA levels are associated with the severity of various neurological diseases including Parkinson’s disease, Huntington’s disease, attention deficit hyperactivity disorder, and many more [66,67]. Various researchers have been working toward the development of electrochemical aptasensors for the screening of DA in clinical samples. Zhou et al. reported a DA specific DNA based electrochemical aptasensor for simple and label-free detection. A DNA aptamer was tethered on an Au-electrode surface via Au–sulfur bonding. In the reported work, methylene blue (MB) was utilized as the intercalating probe for the sensitive and selective detection of DA. The developed electrochemical aptasensor showed a good linear response toward the DA detection in the range of 5–150 nM/L with a LOD at 1.0 nM. The aptasensor exhibited satisfactory selectivity toward DA in blood samples [68]. In another recently published work by Álvarez-Martos and Ferapontova, specific bio-recognition and electro-analysis of DA in serum was reported by an RNA aptamer immobilized on cysteamine-modified Au-electrodes via the alkanethiol linker. The developed DA aptasensor was able to perform continuous amperometric analysis of DA in 10% serum up to 20 h and the detected range of DA was within the physiologically important (0.1–1 μM) range. Reported LOD for DA detection in flow injection mode was 62 nM in the presence of other neurotransmitters (NT) including catechol and L-DOPA. The reported design of the aptasensor utilized both the aptamer alkanethiol tethering to the electrode and screening of the catecholamine-aptamer electrostatic interactions for direct monitoring of DA levels in biological fluids in the presence of competitive NT may be further applied in biomedical research [69]. Furthermore, an electrochemical aptasensor dedicated to DA detection was also developed by Jarczewska et al. The aptasensor was developed by simple tethering of a 50-thiolated 57-mer DNA aptamer probe to the gold electrode. DA concentration was determined without the use of any external redox indicator, but based on the DA oxidation current. The aptasensor exhibited a good linear range from 0.05 to 1 mM/L and LOD of the aptasensor was determined at 26.8 µM/L. The developed aptasensor performance was further improved by using reduced graphene oxide and AuNPs on a glassy carbon electrode, which was followed by the deposition of a thiol-modified DNA aptamer. This modification showed a good improvement in linear response from 5 to 75 mM/L with a LOD at 3.36 mM/L toward the selective detection of DA [58].

The Tau protein is a very important biomarker and plays a crucial role in the pathogenesis of AD. There are very limited reports available for the detection of Tau protein using aptamer-based biosensors. However, very recently, Shui et al. reported a novel aptamer–antibody sandwich assay based on an electrochemical biosensor for the detection of tau-381 in human serum. To improve the detection sensitivity, the aptamer-antibody sandwich assay for the detection of tau-381 was developed by using a tau antibody (anti-tau) and an aptamer specific to tau-381 as the recognition element and cysteamine-stabilized AuNPs for signal amplification. DPV was utilized to record electrochemical signals of tau-381 with different concentrations. The sensor exhibited a wide range of detection between 0.5 pM to 100 pM with an excellent LOD of 0.42 pM for tau-381. The practicability and consistency of the aptasensor were demonstrated by testing tau-381 protein in real patient samples suffering from AD. Authors have claimed that the developed method could possibly be utilized for early-stage screening of AD [70].

Utilization of DNA or RNA based aptamers as bio-recognition elements facilitated the detection of various biomarkers at significantly low concentrations using electrochemical platforms. Furthermore, a considerable improvement of the analytical parameters of electrochemical aptasensors was achieved by utilizing nanomaterials. As most of the aptamers have long strands, a suitable immobilization process must be utilized to achieve efficient binding of the bio-receptor to target biomarkers. To develop advanced and practical utilization of electrochemical aptasensors in clinical biomarker detection, researchers must focus on the miniaturization of aptamer sequences for better tethering and possible applications as a standard diagnostic tool. A tabulated summary is also illustrated with some important biomarkers detected using electrochemical aptasensors. Some of them have not been included in the text, hence can be referred from Table 1.

## 3. Electrochemical Aptasensors for Environmental Sample Application

The present times have observed an increasing demand for the monitoring of natural and hazardous contaminants in the surroundings and maintaining strict control over them. Air, water, and food are the major environmental components getting regularly exposed to several natural and unwanted hazardous chemical contaminants. These contaminants might cause critical health issues and illnesses on living beings [38]. It is consequently critical to developing advanced analytical methods to recognize and enumerate the hazardous contaminants at very low levels. Advancements in electrochemical biosensor platforms were intended to develop rapid, economical, sensitive, and robust methods for easy operations in environmental sample analysis. Such techniques offer better analytical performance without causing much impact on the environment. The discovery of aptamers has provided revolutionary changes in biosensor development [74]. Aptamer-based electrochemical biosensors could surpass conventional analytical methods to some extent for environmental monitoring. Heavy metals and pesticides are the most commonly studied environmental contaminants by various researchers. In recent years, several researchers have reported various electrochemical aptasensors for the analytes related to environmental monitoring. Most representative examples of such methods are elaborated here.

### 3.1. Aptasensors for Heavy Metal Detection

Heavy metal pollution and accumulation in natural surroundings is a great concern due to growing industrialization. Since heavy metal ions are non-biodegradable, they are categorized as a critical cause to contaminate the natural environment around the globe and are responsible for several physiological disorders and illnesses. In recent years, electrochemical biosensors are setting up a new method utilizing aptamers as bio-receptors for the screening of heavy metals in ecological samples. Among the other heavy metals commonly found in environmental commodities, mercury (Hg) is the most common analyte studied extensively. Recently, an aptasensor was developed toward the selective detection of Hg^2+^ ions. Reported aptasensor utilized thymine-Hg^2+^-thymine (T–Hg^2+^–T) chemistry along with nanoporous Au particles for signal amplification. Developed aptasensor was able to detect Hg^2+^ between 0.01–5000 nM, with a LOD at 0.0036 nM. Authors have successfully applied a developed aptasensor for the detection of Hg^2+^ in water samples and claimed a potential perspective for the on-site detection of Hg^2+^ in water samples [75]. In some other reported aptasensors for the detection of Hg^2+^, authors have utilized photoelectric methods along with electrochemical techniques. For instance, Han et al. developed N-doped titanium oxide (TiO_2_) based visible light-activated photoelectrochemical (PEC) aptasensor for label-free determination of Hg^2+^ via quenching of photogenerated electrons [76]. Similarly, Li et al. for the first time fabricated an ultrasensitive and selective PEC aptasensor for detecting Hg^2+^ ions within femtomole (fM) levels by exploring the quercetin-copper (II) complex as the DNA intercalator [77]. In another work, an enzymatic and label-free electrochemical aptasensor was developed for the low-level recognition of lead ions (Pb^2+^). The reported method utilized a metal-organic framework loaded with Ag-Pt particles, which acted as electro-catalytic enhancers to achieve a LOD of 0.032 pM [78]. Similarly, a label-free electrochemical aptasensor for Pb^2+^ was also constructed by Gao et al. using thionine (TH) as the signaling molecule and graphene (GR) as the signal-enhancing platform. The aptasensor was constructed by immobilizing lead specific aptamers on the GR and TH modified electrode. Upon interaction with Pb^2+^, the aptamer probe undergoes a conformational switch from a single-stranded DNA to the G-quadruplex structure, causing the GR with assembled TH to be released from the electrode surface into solution. This reaction resulted in reducing the electrochemical signal of TH on the aptasensor surface. Within the optimized circumstances, the developed aptasensor exhibited a wide linear range for the logarithmic concentration of Pb^2+^ between 1.6 × 10^−13^–1.6 × 10^−10^ M. The LOD of the developed aptasensor was estimated to be at 3.2 × 10^−14^ M. Authors claimed that the developed aptasensor exhibited good reproducibility and selectivity with potential application toward the detection of Pb^2+^ in environmental samples [79].

### 3.2. Aptasensors for Pesticide Detection

Among the various other pesticides, organophosphates (OPs) have gained more attention due to their extremely lethal character and their use in chemical warfare agents. Several investigators have been working toward the development of novel and advanced analytical tools for the screening of such lethal pesticides including methyl-parathion, methyl-paraoxon, chlorpyriphos, and many more. Numerous biomolecules are being utilized for the development of such techniques including antibodies, enzymes, and aptamers. Due to the distinct advantage of substance detection and recognition, several DNA and RNA aptamers are being explored for pesticide screening. However, although several aptamer sequences have already been developed against different pesticides, its full utilization as a bioreceptor in electrochemical biosensor has remained limited. Several researchers have developed different schemes for the development of electrochemical aptasensors toward pesticide sensing. For instance, an impedimetric aptasensor was reported by Madianos et al. for the detection of acetamiprid and atrazine exploring platinum nanoparticles based microwires. Authors have claimed the first time detection of atrazine using an electrochemical aptasensor with a LOD at 1 pM [80]. Recently, an ultrasensitive electrochemical aptasensor was reported for chlorpyrifos. For aptasensor development, chitosan coupled multiwall carbon nanotubes (MWCNTs-CS), along with chitosan functionalized mesoporous carbon and ferrocene hybrid chitosan, were utilized. Under the optimized experimental conditions, the developed aptasensor exhibited a great linear response between 1–105 ng/mL with a reported LOD at 0.33 ng/mL [81]. Acetamiprid is a widely used insecticide that belongs to the neonicotinoid group. Upon leaching into the surroundings, it has severe and toxic effects on living beings. Recently, He et al. reported several aptamers for the selective screening of acetamiprid. The dissociation constant (Kd) of the reported aptamer was 4.98 µM [82]. Furthermore, several other aptasensors have also been developed for the screening of different pesticides. Similarly, Wang et al. reported a DNA based aptamer for multiple organophosphate pesticide detection including phorate, profenofos, isocarbophos, and omethoateas [83]. Several aptasensors have utilized structure switching or the conformational change property of aptamers upon interaction with the target analyte.

Taghdisi et al. [84] developed an acetamiprid electrochemical biosensor, based on the target-induced release of the redox probe methylene blue from the dsDNA formed between the aptamer and complementary strand. Then, methylene blue was detected electrochemically using DPV. Carbendazim, which is employed as a worm control agent, is a widely used, broad-spectrum benzimidazole fungicide. Eissa and co-workers [85] developed a carbendazim electrochemical aptasensor where the aptamer was immobilized onto the gold surface. The specific recognition of carbendazim with the aptamer will lead to conformational changes in the aptamer structure. This conformational change alters the access of a ferrocyanide/ferricyanide redox couple to the aptasensor surface. Then, the response was measured using EIS. Figure 4 represents the schematic for the principle of a structure switching aptamer-based assay. A specific table is presented in Table 2 with some important environmental analytes detected using electrochemical aptasensors.

## 4. Electrochemical Aptasensors for Food Sample Applications

Human health is susceptible to a variety of food-borne substances, which immigrate through the water or food chain. The food-borne substances may be produced naturally or artificially such as natural toxins (mycotoxin), pathogens (microbial), and allergens that are produced naturally. Human development and activities generate a great variety of novel compounds without regard to the consequences of environmental or human health in the long term. Some of them such as pesticides, antibiotics, and food additives are produced artificially. The applications of electrochemical aptasensors in the food sector are mainly focused on the determinations of food-borne substances produced naturally or artificially above-mentioned. Even so, the electrochemical determination principles and forms cannot avoid the one above-mentioned in electrochemical diagnosis applications.

### 4.1. Aptasensors for Food Toxins

Food toxins are normal chemical molecules produced by the metabolic processes of several micro-organisms with injurious effects. Mycotoxins are known as secondary metabolites of fungal origin with severe toxic effects. Plenty of mycotoxins are well categorized with their chemical and physiological properties. Some of the mycotoxins are known to induce genetic diseases and the development of carcinogenic effects [92,93]. Detection of mycotoxins by utilizing aptamers as biorecognition elements in electrochemical biosensors is a prime focused research area. Among the variously reported mycotoxins, ochratoxin A (OTA), aflatoxin M1 (AFM1), and aflatoxin B1 (AFB1) have been widely detected using aptasensors. Aflatoxin B_1_ is a very potent carcinogen produced by *Aspergillus flavus* and *A. parasiticus*. Castillo and co-workers [94] developed an AFB1 sensor that was assembled in a multilayer framework. The Poly (amidoamine) dendrimers of fourth-generation (PAMAM G4) were immobilized onto the gold electrode covered by cystamine and employed for the coupling of single-stranded amino-modified DNA aptamers specific to AFB1. The CV and EIS techniques were used for the detection of AFB1.

Recently, Nguyen et al. reported a CV and SWV based electrochemical aptasensor for AFM1. The aptasensor was constructed by immobilizing AFM1 specific aptamers on interdigitated electrode (IDE) polymerized with Fe_3_O_4_ incorporated polyaniline. The reported aptasensor exhibited good stability, reproducibility, and sensitivity (0.00198 µg/L) toward AFM1 detection. However, the utilization of the developed aptasensor in real sample analysis was not demonstrated [95]. More recently, Mishra et al. have for the first time reported a sensitive detection technique for OTA in cocoa beans via a competitive aptasensor by DPV. In that work, the authors proposed a method where a free and biotin-labeled OTA competed to attach with a tethered aptamer on an SPCE. The detection was performed after adding avidin-alkaline phosphatase (ALP). For detection, the signal was generated via a suitable substrate 1-naphthyl phosphate (1-NP) for ALP. The reported aptasensor exhibited good linearity between 0.15–5 ng/mL with the LOD at 0.07 ng/mL [4]. In another work, Catanante et al. reported a folding mechanism-based aptasensor for OTA detection exploring MB-tagged anti-OTA aptamers. Authors have reported different aptamer coupling techniques using hexamethylenediamine (HMDA), polyethylene glycol, and diazonium coupling. HMDA coupling on SPCE was reported as the best coupling method with LOD at 0.01 ng/mL [96]. A label-free electrochemical impedimetric aptasensor was also developed for OTA detection in cocoa beans. The sensor relies on the specific recognition by the aptamer covalently-bound as a compact monolayer on screen-printed carbon electrodes via the diazonium coupling reaction [5]. Similarly, Gaud et al. recently developed an impedimetric electrochemical aptasensor for the label-free detection of AFB1 in alcoholic beverages. Authors have reported a comparative analysis of two aptamer sequences, namely sequence-A and sequence-B. In the reported work, covalently-bound aptamers as a compact monolayer on SPCE via diazonium coupling allowed the specific recognition of AFB1. A quantitative dynamic range between 0.125–16 ng/mL was reported using EIS for both sequences with LODs at 0.12 ng/mL and 0.25 ng/mL for sequence-A and sequence-B, respectively. Authors have demonstrated AFB1 detection in beer and wine samples to showcase the applicability of the developed aptasensor [35]. Several aptasensors have utilized aptamer sequences along with redox probe and enzyme-based catalytic reactions for food toxin detection. The ricin is another highly potent toxin (a carbohydrate-binding protein) produced by the seeds of the castor oil plant. Recently, Fetter and colleagues [97] have developed a ricin biosensor by coupling an aptamer on the gold electrode surface. Then, the electrochemical signal of labeled redox probe methylene blue was measured using the SWV method for the determination of ricin and botulinum neurotoxins at the nano level in diluted serum.

Fumonisins B_1_ is the most prevalent member of a toxin family, which is produced by several species of Fusarium molds mainly in maize, wheat, and other cereals. Shi [98] developed an aptasensor with the dual amplification of Au nanoparticles and graphene/thionine nanocomposites for the determination of fumonisins B_1_ (FB1). After the specific combination between the aptamer and its target (FB1) in solution, graphene/thionine nanocomposites were released from the electrode surface, resulting in a decreased electrochemical signal using the CV method. Figure 5 represents the detection principle for electrochemical aptasensors utilizing a redox probe attached aptamer and enzyme catalysis for food toxins.

### 4.2. Aptasensors for Antibiotic Residues

Increasing prevalence and occurrence of antibiotic residues in environmental samples and subsequent drug resistance in humans and microorganisms have been an alarming signal around the globe. Thus, the early and precise determination of antibiotics in environmental commodities has gathered considerable focus. Electrochemical aptasensors offer high advantages toward sensitivity and selectivity in the signal measurement. Recent advancements in sensor construction have also enabled them for compatibility with novel microfabrication techniques along with the portability and miniaturization of the sensing platform. Therefore, electrochemical aptasensors have been extremely attractive for several researchers to detect aptamer recognition events in a simple, rapid, and economical manner. The growing presence of antibiotic residues in animal-derived food such as milk and meat has triggered enormous attention toward the progress of rapid and precise analytical techniques for the detection of antibiotics, employing electrochemical aptasensors. Therefore, in recent year, various reports have appeared for antibiotic residue analysis in the food samples. Very recently, an aptasensor was reported for the sensitive detection of streptomycin in milk samples. The sensor utilized an arch shape aptamer and its complementary strand along with exonuclease. The addition of streptomycin induces conformational changes between the aptamer-streptomycin conjugate and release a complementary strand whereas exonuclease acts as a digestive enzyme that particularly degrades the ssDNA from its 3′-terminus end. The addition of exonuclease causes the degradation of the complementary strand and resulted in the generation of electrochemical signals. The reported aptasensor was highly specific for streptomycin detection with a LOD at 11.4 nM. Furthermore, the authors extended this aptasensor for streptomycin detection in serum and were able to achieve a LOD as low as 15.3 nM. The aptasensor exhibited excellent linear ranges for streptomycin in serum (40–1500 nM), and milk (30–1500 nM) [99]. Zhou et al. reported an aptasensor for label-free determination of kanamycin in milk. The authors deployed the SWV technique to quantify kanamycin residues via a current response that resulted through the conformational change of a specific aptamer coupled on an Au-electrode. The developed aptasensor exhibited an excellent range of detection between 10–2000 nM [100]. Similarly, in other reported work by Liu et al. a sandwich-type electrochemical aptasensor was developed for the determination of oxytetracycline residues. The aptasensor construction was based on the graphene-three dimensional nanostructure gold nanocomposite (GR3DAu), which explored an aptamer–AuNPs–horseradish peroxidase nanoprobe for signal amplification. The developed aptasensor exhibited an excellent linear range for oxytetracycline detection between 5 × 10^−10^–2 × 10^−3^ g/L, with a LOD at 4.98 × 10^−10^ g/L [101]. Kim et al. reported a tetracycline specific aptasensor utilizing the ssDNA aptamer as the bioreceptor. The CV and SWV techniques were employed to detect the binding of biotinylated ssDNA aptamer immobilized on a streptavidin modified SP-Au-electrode and tetracycline. The reported aptasensor exhibited excellent sensitivity toward tetracycline detection with a wide range between 0.01–10 μM. Authors have claimed the applicability of the developed aptasensor in food samples for low- and high-level detection of tetracycline [102]. Furthermore, another aptasensor based on a single-stranded DNA-binding protein was recently reported for the ultrasensitive detection of ciprofloxacin. The developed aptasensor exploited a single-stranded DNA-binding protein (SSB) on an Au-electrode and demonstrated high specificity toward ciprofloxacin determination in milk samples with an excellent LOD at 263 pM [103]. Some other aptasensors have also been reported on the same analyte by integrating nanomaterials and nanocomposites [104,105]. The majority of developed aptasensors have demonstrated a highly suitable detection range for the quantification of sensitive and specific analysis for different antibiotics in various food matrices. Most of the reported aptasensors have claimed the potential for industrial applications. Some of the important electrochemical aptasensors are presented here in Table 3.

## 5. Conclusions and Future Perspectives

Biosensors have been recognized as a distinct analytical tool for the rapid and sensitive detection of clinically important biomarkers, lethal and hazardous chemicals, food toxins, and microbial contaminants. Among the other reported biosensors, electrochemical biosensors offer various advantages, particularly easy miniaturization of the detection platforms. This review illustrates the most representative reports on electrochemical aptasensors dedicated to clinical diagnostics, food, and environmental monitoring as reported in recent years. The application of aptamers as bio-recognition elements is profitable, as they demonstrate a higher binding affinity for the target analyte. Even though the described aptasensors are distinguished by their ability to detect low levels of target analytes, there are several issues to be resolved during their development. First and foremost, choosing a specific aptamer sequence is a tedious task since the aptamer selection process needs to be adjusted as per the target analytes. Second, immobilization of the aptamer probe on a sensing electrode remains a laborious task. Immobilization of the aptamer sequence should be selected wisely to provide competent binding of the target analyte with specific aptamers. The integration of functional nanomaterials and nanocomposites could contribute to enlarge the sensing surface and thus improve the conductivity of the sensing electrode; however, it usually involves multiple steps of surface modification. Furthermore, most of the developed aptasensors work by exploring redox probes. Thus, the redox probe must also be wisely selected as per the experimental parameters. Recent developments in electrochemical aptasensors have effectively reduced the time consumption associated with conventional lab techniques by providing multi-analyte and high-throughput sensing platforms for clinical, food, and environmental applications. The development of electrochemical aptasensors for the concurrent screening of multi-analytes seems difficult since it will be very complex to immobilize multiple aptamers on a single sensing platform. However, this concern could be addressed by designing an engineered aptamer sequence for specific binding to several targets with a similar affinity. Moreover, it can be foreseen that the development of new structure switching aptamers for the specific recognition of various clinically important biomarkers and hazardous chemicals will come up in the near future. Integration of lab-on-chip platforms with aptamers and functional nanomaterials will open new avenues for the improvement of handy and point-of-care devices for clinical diagnostics, food analysis, and environmental monitoring.

## Figures and Tables

**Figure 1 sensors-19-05435-f001:**
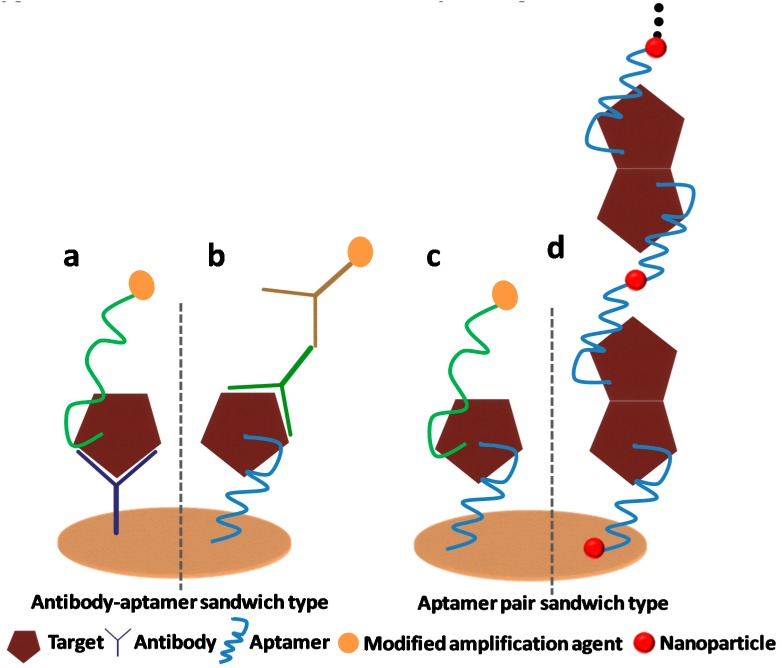
The schematic illustration of various assays combining antibody and aptamer coupling on a sensor surface. (**a**) Antibody and aptamer originated sandwich bio-assay, (**b**) aptamer and antibody-bioassay, (**c**) binary aptamer originated sandwich-bioassay, (**d**) aptamer-based sandwich-bioassay based on smart nanomaterials.

**Figure 2 sensors-19-05435-f002:**
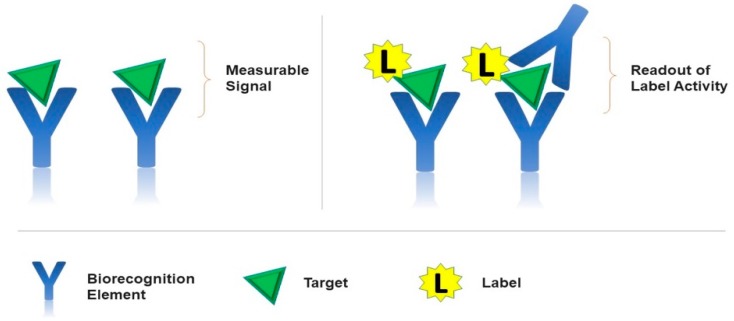
Schematic representation of labeled and label-free approaches in electrochemical biosensors.

**Figure 3 sensors-19-05435-f003:**
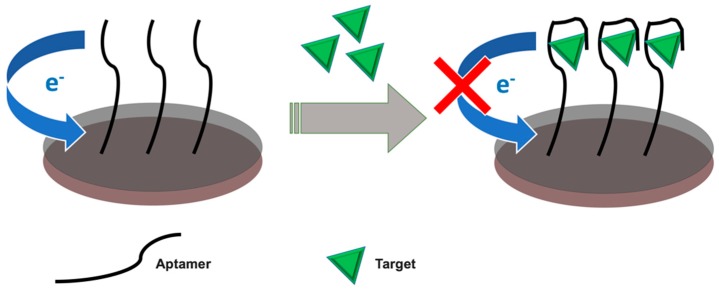
Schematic representation of target-induced variation in charge transfer resistance.

**Figure 4 sensors-19-05435-f004:**
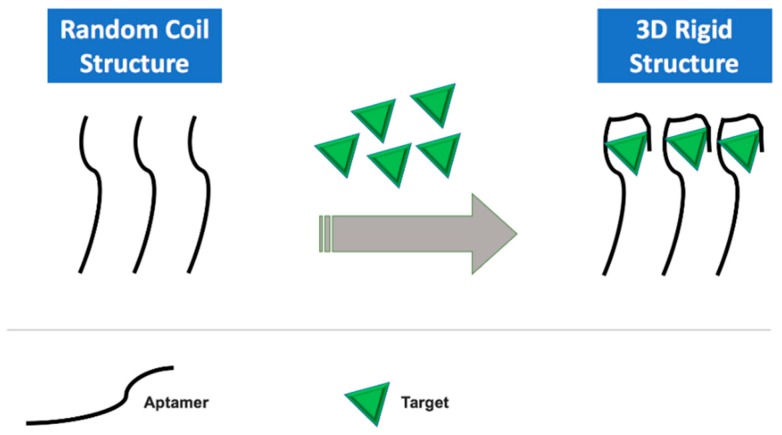
Schematic for the principle of the structure switching aptamer-based assay.

**Figure 5 sensors-19-05435-f005:**
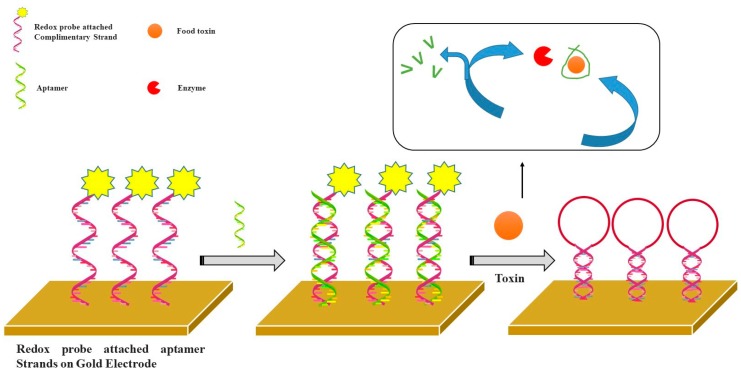
Schematic principle of the electrochemical aptasensor for food toxin detection via the redox probe attached aptamer and enzyme induced catalysis assay.

**Table 1 sensors-19-05435-t001:** Important analytes (biomarkers) detected using electrochemical aptasensors for clinical diagnostic applications.

S.N	Analyte	Detection Method	LOD/Range	System	Reference
1	PSA	EIS	1 ng/mL	Thiol terminated sulfo-betaine	[52]
2	PSA	DPV	0.25 ng mL^−1^	Graphitized meso-porous carbon nanoparticles	[53]
3	OPN	SWV	0.4–4.5 nM	Gold/DPA	[59]
4	OPN	SWV	3.7 nM	Biotinylated RNA aptamer	[60]
5	CRP	DPV	0.2 mg/L	Mangnetic nanoparticles on SPCE	[62]
6	CRP	SWV	0.0017 ng mL^−1^	Functionalized silica–Au nanoparticles	[64]
7	IL-6	CV/EIS	0.33 pg mL^−1^	SPCE-polypyrole	[65]
8	Tau-381	DPV	0.42 pM	Cysteamine-stabilized gold nanoparticles (AuNPs)	[70]
9	CRP	SWV/EIS	100 pM	Thiolated aptamers-Au surface	[71]
10	PSA	Potentio-amperometric	0.064 pg mL^−1^	Functionalized graphene-modified carbon screen-printed electrodes as tr	[72]
11	PSA	DPV	28 pg/mL	Ag/CdO nanoparticles-graphene oxide nanosheet	[73]

**Table 2 sensors-19-05435-t002:** Important analytes detected in environmental matrix using electrochemical aptasensors.

SN	Analyte	Detection Method	LOD/Range	System	Reference
1	Hg^2+^	DPV	0.0036 nM	Thymine- Hg^2+^ -Thymine	[75]
2	Hg^2+^	Photoelectrochemical	2–6 µM	N-doped-TiO_2_	[76]
3	Hg^2+^	Photoelectrochemical	3.33 fmol/L	PCTA/GO	[77]
4	Pb^2+^	DPV	0.032 pM	Ag/Pt nanoparticle	[78]
5	Pb^2+^	DPV	3.2 × 10^−14^ M	Graphene/Thionine	[79]
6	Pb^2+^	EIS	1.67 pmol/L	Au@p-rGO	[86]
7	Pb^2+^	SWV/EIS	32 pM	hemin/G-quadruplex -based DNAzym	[87]
8	Profenofos, Phorate, Isocarbophos, Omethoate	DPV	0.003 nM, 0.3 nM, 0.03 nM and 0.3 nM	GO-CuNPs	[88]
9	Pb^2+^		312 pM	AuNPs- hairpin-aptamer and thionine.	[89]
10	Hg^2+^	EIS	0.005 ppm	ink-jet printed gold electrodes	[90]
11	Acetamiprid	DPV	153 pM	SiNP-streptavidin conjugate modified MB-dsDNA	[84]
12	Acetamiprid	EIS	3.3 × 10^−14^	Ag nanoparticle decorated nitrogen doped graphene	[91]

**Table 3 sensors-19-05435-t003:** Aptasensors for food toxins detected by electrochemical techniques.

S.N	Analyte	Detection Method	LOD/Range	System	Reference
1	AFB1	CV/EIS	0.03 nM	Poly(amidoamine) dendrimers	[94]
2	AFM1	SWV	1.98 ng·L^−1^	Polyaniline (Fe3O4/PANi)film	[95]
3	OTA	DPV	0.01 ng/mL	HMDA-MB system	[96]
4	FB1	CV	1 pg/mL	AuNPs)and graphene/thionine nanocomposites	[97]
5	Streptomycin	CV/DPV	11.4 nM	Aptamer-on gold electrode	[99]
6	Kanamycin	SWV	10–2000 nM	Aptamer-on gold electrode	[100]
7	Oxytetracycline		4.98 × 10^−10^ g L^−1^	Graphene three dimensional nanostructure gold nanocomposite	[101]
8	Tetracycline	SWV	10 nM	Streptavidin-modified screen-printed gold electrode	[102]
9	Ciprofloxacin	EIS	0.5 ng mL^−1^	CNT- V2O5-chitosan	[104]
10	FB1	EIS	2 pM	Thiolated aptamers on AuNP	[106]
11	OTA	EIS	0.15 ng/m	Di-azonium coupled reaction	[5]
12	OTA	DPV	0.07 ng/mL	APL-pNPP based	[4]
13	OTA /FB1	SWV	10 pg mL^−1^ to 10 ng mL^−1^ and 50 pg mL^−1^ to 50 ng mL^−1^	Magneto-controlled aptasensor	[107]
14	AB1	DPV	0.002 fg/mL	Reduced graphene oxide/molybdenum disulfide/polyaniline@gold nanopa	[108]
15	OTA	CV	0.05 nM	Gold electrode covered with electropolymerized neutral red and silver nanoparticles	[109]
16	AFB1	CV/EIS	0.1 nM and 0.05 nM	Glassy carbon electrodes modified with electropolymerized Neutral red and polycarboxylated macrocyclic ligands	[110]
17	OTA	EIS/CV	0.1 ng/mL in	A Langmuir–Blodgett (polyaniline (PANI)–stearic acid (SA)) film	[111]
18	Zearalenone	CV	0.17 pg mL	Molybdenum disulfide (MoS2) doped multi-walled carbon nanotubes (PEI-MoS2-MWCNTs) nanohybrid	[112]

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
