# Peer review of "Application of Electrochemical Aptasensors toward Clinical Diagnostics, Food, and Environmental Monitoring: Review"

_sensors, 2019, doi:10.3390/s19245435_

Round 1
Reviewer 1 Report
The review by Z. Li et al aims to describe aptamer detection for electrochemical approaches essentially concentrating on three applications. There are some points that could be included/modified to further improve the quality of the manuscript. I would therefore consider it favorable for publication in Sensors under the provision that the following concerns are adequately addressed.
(1) Many relevant reviews of electrochemical aptasensors from the past five years have been published, and thus what is the purpose of your review and what is the difference between yours and other reviews? In addition, the introduction is far too general and vacuous, and needs to be written more concisely with more focus on the application issue.
(2) Please rewrite the abstract, it needs more elaboration to dictate the theme of the review article.
(3) The author did not put enough efforts on the presentation of figures. For example, Figure 1(e) made me confused because the topic of this review is about the electrochemical detection, and thus the illustration of FRET detection (Figure 1e) should be not found in this manuscript. Figure 1(d) showed the aptamer-based sandwich-bioassay based on smart nanomaterials. However, only sensor surface could be found, and no nanomaterials were shown in Figure 1(d). Figure 2, 3, and inset of Fig.4 (should be 3, 4, and inset of Fig. 5) were not well rendered. Poorly rendered figures might confuse the readers or clutter the manuscript. Please revise your figures seriously.
(4) The technical descriptions are too light or even missing. For example, line 249, “…This developed aptamer showed a good response towards OPN with detection limit of 1.4 nM with good reproducibility and acceptable selectivity [59]. Another work on Osteopontin was developed by the same group. They developed a label-free electrochemical RNA aptasensor to detect human osteopontin with the LOD of 3.7 nM [60]….” Why did the same group report the similar studies and what is the difference between both? The author should discuss them in the manuscript.
(5)The authors reported a lot of studies in three applications which overload the text with information about sensitivity, detection limit, and sensing manner in the manuscript. They should consolidate similar application information into one table to let the readers compare easily so that the readers does not have to search for related information. Thus, please create three tables which summarise the sensing performance of aptasensors in these three applications.
Whilst overall a fairly representative description is provided, the general result is very mosaic because of the disparity in both scope, presentation and description of the various sections.
Minor comments
(1) line 146, it should be Figure 2.
(2) line 718, it should be reference 51.
(3) The format of references should be consistent. Journal names are abbreviated and italicized. DOI(doi:10.1038/xxxxx.) should be given in every reference.
Author Response
Thank you for your valuable suggestions. Please see the attachment.

Reviewer 2 Report
The review article by Li et al. describes recent advances in electrochemical sensing of relevant molecules/proteins in clinical and food/environment diagnostic using aptamers as main recognition element. There a lot of reviews on this field published in literature even in the last 4 to 5 years, also in mdpi journals. This review gives a well-prepared insight into new published data starting with chapter 2. However, the first part must be reworked in terms of completeness and structured more precisely. The work is worth publishing after major revisions.
Following parts need to be addressed:
Chapter 1.1: You refer to design of aptamer-based sensors (also including) AB in part a) and then switch to EIS as measurement method in part b). This makes no sense to the reader. I suggest a first chapter on all possible design of aptamer sensors used for electrochemical measurement (also including the pictures in the later part. Figure 2/3/4). Afterwards a chapter with all electrochemical techniques used for the referenced study should be stated not only EIS. You already state those techniques in line 159ff. There should also be citations, where these techniques are used (only 2 in the EIS chapter!) A table summing up all techniques and measurement principles with references would help the reader a lot.
Figures should ne reworked in quality and may give a little more detail (e.g. Figure 2).
Check for your SI units: M stands for Molar, it’s a concentration unit mol/L. So, there cannot be a M/L unit! Also, there are some cases like line 534 where you use 10-10 g/l. Use here the SI prefixes like you did in the other cases pg/ng etc.
Please include an abbreviation List: As there a tons of abbreviations its always complicated to look for them in the text
Minor changes:
Figure enumeration is wrong: you have two times Figure 1. Line 71: field instead of filed Line 77: Different format than line 122 Line 87: check the sentence Line 105/6: -1 must be superscripted Line 107: l in Lysozyme small Line 227: spacing missing before (MPA) Line 472: spacing missing after 0.01 Line 475 and 534: point missing after et al
Author Response

(The authors gave the same response as above.)

Round 2
Reviewer 1 Report
I am very happy to see that revised manuscript has improved significantly in response to the reviewer comments, although I still find that English should be improved in the entire manuscript. Especially of the newly written parts. For example,
Line 37 Although, aptamers have various other applications, but …
Line 288 Another work on Osteopontin was developed by the same group.
Line 291 What does the letter “rhOPN” stand for? recombinant human osteopontin?
Line 293 …analyte-aptamer interaction as well as binfing (binding?)
Line 388,391,484,605 electrochemical Aptasensors
Reviewer 2 Report
The authors answered to all concerns. The review can be published in the present form.
Author Response
Thank you for your recommendations on the manuscript.